# HAPNEST: An efficient tool for generating large-scale genetics datasets from limited training data

**Sophie Wharrie**[1]    **Zhiyu Yang**[2]    **Vishnu Raj**[1]    **Remo Monti**[3]    **Rahul Gupta**[4]
**Ying Wang**[4]    **Alicia Martin**[4]    **Luke J O'Connor**[4]    **Samuel Kaski**[1,5]
**Pekka Marttinen**[1]    **Pier Francesco Palamara**[6]    **Christoph Lippert**[3,7]    **Andrea Ganna**[2,4]

[1]Aalto University    [2]University of Helsinki    [3]Hasso Plattner Institute for Digital Engineering, University of Potsdam    [4] Broad Institute of MIT and Harvard    [5] University of Manchester    [6] University of Oxford    [7] Hasso Plattner Institute for Digital Health at Mount Sinai, Icahn School of Medicine at Mount Sinai

{sophie.wharrie,vishnu.raj,samuel.kaski,pekka.marttinen}@aalto.fi
{zhiyu.yang}@helsinki.fi {remo.monti,christoph.lippert}@hpi.de
{rahul,yiwang,armartin,loconnor,aganna}@broadinstitute.org
palamara@stats.ox.ac.uk

## Abstract

In this extended abstract we present a new highly efficient software tool called HAPNEST that enables machine learning practitioners to easily generate and evaluate large synthetic datasets for human genetics applications. HAPNEST enables the generation of diverse synthetic datasets from small, publicly accessible reference datasets. We demonstrate the suitability of HAPNEST-generated data for supervised tasks such as genetic risk scoring. The HAPNEST software can be accessed at `https://github.com/intervene-EU-H2020/synthetic_data`.

## 1   Introduction

Machine learning applications related to human genetics require large datasets that fairly represent the diversity of the human genome, yet privacy concerns and other practical constraints make it difficult for machine learning practitioners to acquire this data. Synthetic data is a promising alternative, but for privacy reasons, training data is limited to small, publicly available datasets. Furthermore, existing reference-based simulation methods [1, 2, 3] do not easily scale to generating large (> 1 million individuals) synthetic datasets. Thus new scalable approaches are required to generate large, high-fidelity synthetic datasets that generalize beyond the small number of training examples.

In this extended abstract we present a new method called HAPNEST that enables efficient, large-scale generation of genetics and phenotypic data for multiple ancestry groups. In addition to synthetic data generation tools, the containerized HAPNEST software application also includes tools for evaluating synthetic data quality, from the perspectives of fidelity, generalizability and diversity. We demonstrate that an approximate Bayesian computation (ABC) procedure formulated to give preference to models generating high-fidelity synthetic samples that have low genetic relatedness with the reference data results in synthetic data that preserves key statistical properties of real genetics data with better generalization than alternative approaches.

Our intention is for HAPNEST-generated data to be used as a benchmark for supervised learning tasks with common genetic variants and complex disease traits. We present an application demonstrating the utility of HAPNEST-generated data for predicting an individual's genetic risk of disease.

NeurIPS 2022 Workshop on Synthetic Data for Empowering ML Research.

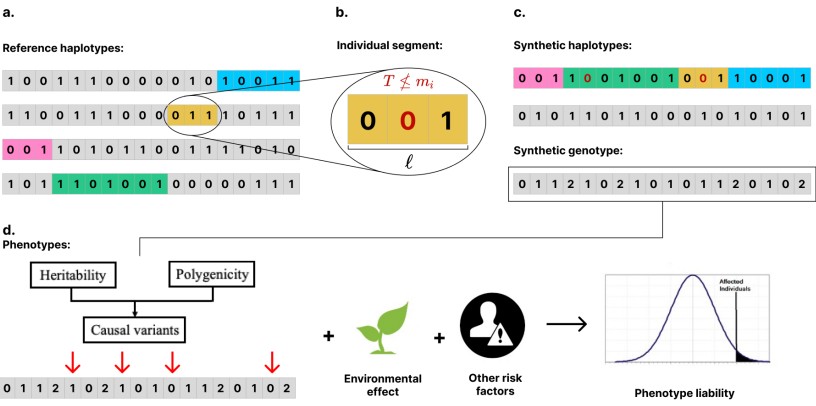

Figure 1: A set of real haplotypes (**a.**) are imperfectly copied segment-by-segment (example of an individual segment in **b.**) to construct a synthetic haplotype (**c.**). A phenotype is simulated as a summation of genetic, covariate and environmental (noise) effects (**d.**).

## 2 HAPNEST

Previous approaches to generating synthetic data in genetics include coalescence-based approaches [4], which simulate artificial genomes using predefined statistical models, and reference-based approaches [1, 2, 3], which construct artificial genomes by resampling segments from a reference dataset. Our approach builds upon ideas from this second class of methods, which are known to preserve key statistical properties (e.g. linkage disequilibrium patterns) of real genetics data for common genetic variants, which form the bulk of complex disease heritability.

HAPNEST generates synthetic data of the form $(X^{(i)}, Y^{(i)})_{i=1}^{n}$ for $n$ synthetic individuals, where $X^{(i)} \in \{0, 1, 2\}^m$ represents a genotype for $m$ common genetic variants (i.e. single nucleotide polymorphisms) and $Y^{(i)} \in \mathbb{R}^k$ represents $k$ (binary or continuous) phenotypic traits. A genotype $X^{(i)}$ is composed of two haplotypes, $(H_1^{(i)}, H_2^{(i)})$, inherited from two parents, such that $X^{(i)} = H_1^{(i)} + H_2^{(i)}$.

A synthetic haplotype is created by imperfectly copying segments of various lengths from reference haplotypes belonging to a certain ancestry group $s$ (Fig. 1a) (alternatively, a weighted combination of multiple ancestry groups can be used, but caution must be taken when interpreting admixed samples). The segment length $\ell$ (in centimorgans) is sampled according to a stochastic model that simplifies coalescent and recombination processes for two haploid individuals,

$$\ell \sim Exp(2T\rho_s), \ T \sim Gamma(2, N_s^{ref}/N_{e,s}), \tag{1}$$

where $T$ represents coalescence time, $\rho_s$ is the population-specific recombination rate, $N_{e,s}$ is the population-specific mean effective population size, and $N_s^{ref}$ is the number of reference samples for population $s$. To aid generalizability, segments are imperfectly copied by only copying a genetic variant at position $j$ if $T \leq \mu_j$, where $\mu_j$ is the variant's age of mutation (obtained from [5]) (Fig. 1b). Individual segments are copied in this manner until all variant positions are filled (Fig. 1c). Overall, this results in a scalable algorithm for which we have developed a highly efficient multi-threaded and parallelizable implementation in the Julia programming language.

The populations $S = \{AFR, AMR, CSA, EAS, EUR, MID\}$ represent 6 major continental ancestry groups typically recognized in human genetics research, for which sufficient reference data was available from the publicly accessible 1000 Genomes Project and HGDP datasets [6]. Since genetic reference datasets are typically over-represented by European-ancestry individuals, to avoid biases against underrepresented groups we model these 6 populations separately using an approximate Bayesian computation (ABC) procedure (described in the following section).

Once the genotypes have been created, HAPNEST assigns phenotypes to each individual using the liability threshold model. Standardized phenotypic liability for each sample is simulated as a summation of genetic, covariate and environmental effects, where the genetic effect is generated via

an additive manner from effects of all phenotype causal variants (Fig. 1d). Users can specify the variance contribution of each liability component. Further, for the genetic component, users can also define the number of causal variants, or input a list of designated variants to be causal. Effect size $\beta$ of each causal genetic variant $i$ follows a Gaussian mixture distribution:

$$\beta_i \sim \sum_{j=1}^{k} \pi_j N(0, \sigma_j^2), \quad \sum_{j=1}^{k} \pi_j = 1,$$

where $k$ is the number of components and $\sigma_j^2$ is the distribution variance for the $j-$th component. Each SNP falls into one of the mixture components $j$ with probability $\pi_j$. All parameters can be customized by the user. HAPNEST also allows simultaneous generation of multiple genetically correlated phenotypes, and correlated effect sizes across different ancestry groups. Once the liability is generated, patient/healthy control statues can subsequently be assigned to an individual based on a threshold. Individuals with a liability value above the threshold will be considered patients, whereas the rest will be considered as healthy controls.

## 3 Results

### 3.1 Population-wise parameter tuning using ABC

We use approximate Bayesian computation (ABC) rejection sampling to estimate the posterior distributions of the unknown model parameters, $\rho_s$ and $N_{e,s}$, for each population $s$. Preference is given to synthetic datasets that preserve the linkage disequilibrium (LD) structure of the reference data (fidelity objective), while limiting the genetic relatedness between the synthetic data and the real data (generalizability objective). This is achieved by giving high probability to parameter configurations that minimize the discrepancy $d(Q(\mathcal{D}_{real}), Q(\mathcal{D}_{synthetic}))$, where $d$ is Euclidean distance and $Q$ is a vector concatenating LD decay and cross-relatedness (the proportion of first degree and second degree relatives, between the real and synthetic datasets for $\mathcal{D}_{synthetic}$ and between two random partitions of the real dataset for $\mathcal{D}_{real}$). For computationally expensive simulations of large synthetic datasets, HAPNEST trains a Gaussian process regression model using a small number of simulations to estimate the remaining simulation results [7].

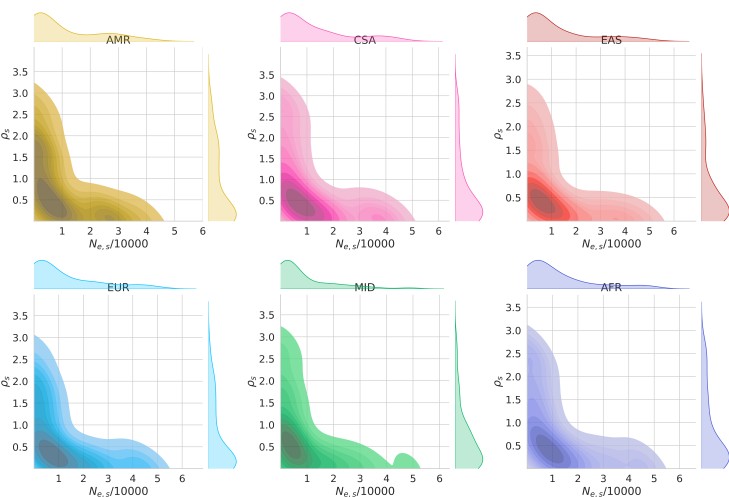

Figure 2: Posterior distributions plotted as marginal and bivariate kernel density estimates for the unknown model parameters, $\rho_s$ and $N_{e,s}$, for each population $s$. The experiment setup used 500 simulations for 1000 synthetic samples based on a reference of chromosome 21 HapMap3 variants, with uniform priors and a 20 percent rejection rate.

The posterior distribution of parameters estimated by ABC for each of the six sub populations is given in Fig. 2. These parameter configurations preserve LD structure between the real and synthetic datasets, which is important for downstream applications, while limiting the degree of cross-relatedness to encourage generalization of the synthetic samples.

## 3.2 Evaluation

To evaluate the quality of data generated by the synthetic data generation algorithm, we consider three key aspects of the data for downstream tasks. For each of these aspects, we design and evaluate multiple metrics to analyze the robustness of the synthetic data generation algorithm.

1. Fildelity: Realistic synthetic genetic data should retain the key summary statistics of the original genetic data.

2. Diversity: Synthetic genetic data should be sufficiently different from the real-world genetic data, to preserve privacy.

3. Validity: The synthetic phenotypes should behave similarly to the real-world phenotypes in downstream genetics applications.

We also benchmarked the phenotypes' heritability and genetic correlation of the synthetic dataset to ensure the validity of phenotype generation using widely used genomic analyses tools CGTA-GREML [8] and LDSC [9]. We observed high concordance between user-specified parameters and estimates from the tools.

### 3.2.1 Fidelity of synthetic samples

We compare four properties of the synthetic dataset against real dataset to understand the reliability of synthetic samples in retaining key statistical properties.

1. *Minor Allele Frequency.* The discrepancy of Minor Allele Frequency (MAF) from the original dataset is measured using two statistical distances. Considering real data and synthetic data as a collection of realized Bernoulli random variables with different parameters, we can compute statistical discrepancies between these two random variables.

2. *Population structure* Most of the downstream Polygenic Risk Score (PRS) tasks are interested in analyzing the effect of SNPs at a population level, and it is extremely critical to maintain the genetic structure of real population in the synthetic samples. We use the popular concept of Principal Component Analysis to ensure that the synthetic samples are well aligned with the real samples. We compute the cosine similarity between the principal components computed on both real data and synthetic data to understand how well the population structure is retained in the synthetic data.

3. *Linkage Disequilibrium* While measuring MAF preserves the probability of occurrence on each SNP, the co-occurrence of neighboring SNPs are not ensured due to the IID assumption. In order to verify the local structure of co-occurrence is maintained in the synthetic dataset, we compute the Linkage Disequilibrium correlation matrix and compare them between real data and synthetic data.

4. *Nearest Neighbour Adversarial Accuracy* A sufficiently good synthetic data sample must be indistinguishable from the real data sample. Based on this principle, we use the adversarial accuracy $AA_{TS}$ [10] of a classifier to quantify the closeness of the synthetic dataset to the real dataset. When the synthetic data is very close to real data, we will not be able to distinguish real data samples from synthetic samples and hence $AA_{TS} \rightarrow 0.5$.

### 3.2.2 Diversity of synthetic samples

The main aim of creating synthetic data is to preserve the privacy of individual samples in the original dataset yet allowing the scientific community to develop new tools from the vast amount of data available. We study the 'closeness' of synthetic samples to real samples using two key metrics.

1. *Kinship analysis*: For kinship analysis [11], we compute the kinship between samples within the generated dataset as well as kinship across the reference dataset and generated dataset. Within-dataset kinship analysis is useful in analyzing the diversity in the synthetic samples within the cohort and kinship across datasets is useful to address the privacy concerns of reference data leaking into synthetic data as well as quantifying the relatedness between synthetic and real samples.

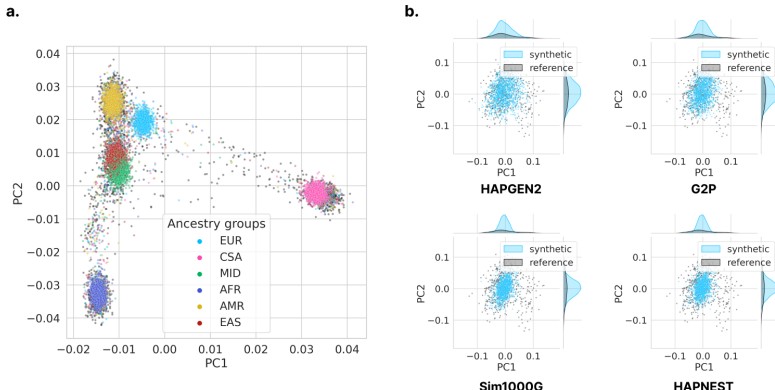

Figure 3: **a.** PCA projection plot for 10,000 multi-ancestry synthetic samples generated by the HAPNEST method, for chromosome 21 HapMap3 variants; **b.** Comparison of PCA projection plots and bivariate densities for 1000 European-ancestry synthetic samples.

2. *IBS analysis* Identical-by-state (IBS) analysis shows samples which share an identical sequence at a particular locus and is useful in analyzing the closeness between samples. We analyze the distribution of IBS values to compare the real data with generated synthetic data.

Table 1: Comparison of evaluation metrics for four synthetic data generation tools. Each synthetic dataset contains 1000 samples, based on a European ancestry reference panel for HapMap3 variants on chromosome 21. $\downarrow$ means lower the better, $\uparrow$ means higher the better. For $AA_{TS}$, a value near 0.5 is better. No time is reported for G2P and Sim1000G as they failed to generate $1M$ samples. HAPNEST results use the mean of the posterior distribution obtained from ABC.

| Metric | HAPGEN2 | G2P | Sim1000G | HAPNEST |
|---|---|---|---|---|
| $AA_{TS}$ | 0.623 | **0.505** | 0.706 | 0.550 |
| LD decay (Euclidean distance)$\downarrow$ | 0.014 | **0.013** | 0.442 | 0.066 |
| MAF (Wasserstein divergence)$\downarrow$ | 0.019 | **0.012** | 0.013 | 0.014 |
| PC alignment$\uparrow$ | **0.311** | 0.222 | 0.043 | 0.192 |
| Related pairs (Within dataset, < 1st degree)$\downarrow$ | 2602 | 3 | 2492 | **0** |
| Unrelated pairs (Within synthetic dataset)$\uparrow$ | 479613 | 487249 | 402587 | **494944** |
| Related pairs (Between real and synthetic)$\downarrow$ | 2031 | 7 | **0** | **0** |
| Unrelated pairs (Between real and synthetic)$\uparrow$ | 731464 | 734478 | 689120 | **742229** |
| Time required ($1M$ samples, in minutes)$\downarrow$ | 173 | - | - | **45** |

The results of comparison are given in Fig. 3 and Table 1. We can observe that HAPNEST is able to provide a balance between fidelity, diversity of samples and speed.

# 4 Application of HAPNEST

Using synthetic genotype and phenotype data generated by HAPNEST, we performed a number of supervised tasks that are common in statistical genetics analyses, including genomewide association study (GWAS) and polygenic risk score (PRS) computation. GWAS is widely used to detect variants that are associated with the phenotype, and GWAS carried out on synthetic genome data can further allow us to compare the association with pre-defined causal variants. As an example, we present in figure 4 a GWAS Manhattan plot for a trait with rather low genetic contribution to liability (heritability) and relatively small number of causal variants (polygenicity). Due to LD, variants that are not causal themselves, but locate nearby the causal ones, are also found associated with the phenotype, indicating HAPNEST well captures linkage structures in the genome.

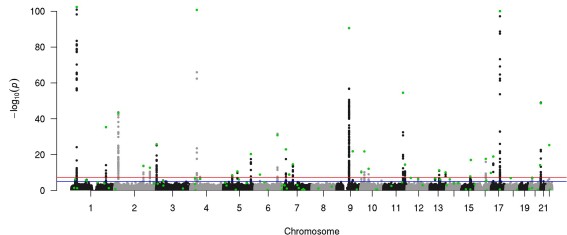

Figure 4: **Example GWAS Manhattan plots.** Typical GWAS results for phenotype with low genetic contribution (heritability = 0.1) and low number of causal variants (polygenicity = 0.0001, ie. approximately 0.01% of total number of variants having causal effects on phenotype liability). Colored in green are dots representing causal variants.

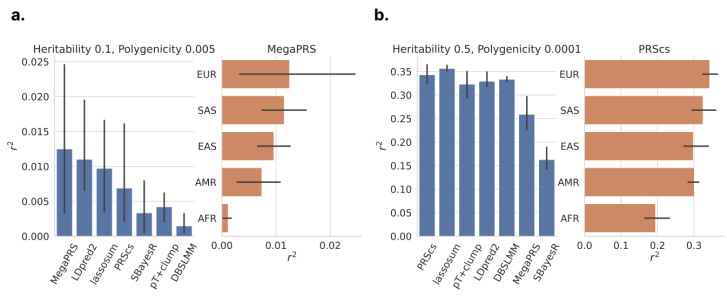

Figure 5: Pearson correlation (squared) between predicted and observed values, for various PRS methods and two European-ancestry phenotypes with varying genetic architectures. For the best method in each case, we show the performance across multiple ancestry groups.

One of the most popular downstream applications of GWAS results is computing PRS, which is an individual level score that characterizes the genetic risk of one being affected by a phenotype of interest. The score is computed as a weighted sum of risky alleles in ones genome. From GWAS, we can identify genetic variants that are associated with a phenotype and their estimated effect sizes, and those effect sizes can then be used to derive weights for genetic scores computation in another cohort. The score, as it in theory captures the genetic risk for an individual, can be viewed as, to certain extent, a predictor of one's phenotype. Its predictive power can by evaluated by regressing the score over samples phenotype and examining the Pearson correlation $r$, $r^2$ or similar statistics for fitness. Here, we applied PRSpipe, a pipeline developed to evaluate 7 existing and widely used PRS scoring methods, on synthetic genotype and phenotype data generated by HAPNEST. We saw similar results as observed previously on real genetic data (Fig. 5), demonstrating the validity of HAPNEST's output.

## 5 Concluding Remarks

In this work we present a new software tool called HAPNEST that efficiently generates and evaluates large synthetic datasets for human genetics applications. Despite the limitations of working with genetics data, such as privacy concerns, we demonstrate that HAPNEST can generate reliable and generalizable synthetic datasets, that are suitable for downstream applications. We hope that the machine learning community will find HAPNEST useful as an aid for methods development for supervised tasks such as genetic risk scoring.

## 6 Acknowledgments

This project has received funding from the European Union's Horizon 2020 research and innovation programme under grant agreement No. 101016775.

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
