# OpenReview forum: "HAPNEST: An efficient tool for generating large-scale genetics datasets from limited training data"
_NeurIPS.cc/2022/Workshop/SyntheticData4ML — Neurips 2022 SyntheticData4ML_

### Official Review · Reviewer_iZwa · 2022-10-17
**How to make sure the generated data is not biased?**

**Rating:** 7
**Confidence:** 3

**Review:**

The authors proposed a tool for efficiently generates and evaluates large synthetic datasets for human genetics applications. The effectiveness of this tool has been evaluated through multiple supervised tasks. The paper is fairly well structured. My main concern with the software is that what is the guarantee to generate data out of observed domain. In other words, how the proposed model is generalizable?

---

### Official Review · Reviewer_7PZJ · 2022-10-18
**Interesting, well thought-through and relevant approach**

**Rating:** 7
**Confidence:** 4

**Review:**

I recommend this extended abstract for the presentation at the workshop.
For extended abstract, the work is thoroughly explained, the approach is sound and all the major areas within this domain (synthetic genomic sequence data) are addressed.

Couple of suggested revisions:

* give more details about the unique contribution of HAPNEST with respect to the related work
* add more details to the abstract
* maybe be a little more specific/descriptive in how HAPNEST specifically aids the (poly) genetic risk score research

---

### Official Review · Reviewer_p18F · 2022-10-18
**HAPNEST: A tool for generating genetics data**

**Rating:** 6
**Confidence:** 2

**Review:**

This work presents a statistical simulation method for generating genetical sequence data that preserves the statistical characteristics of the real world data. They model the sequence length $l$ and coalescence time $T$, as exponential and gamma distributions specifically to represent a combination of two ancestry individuals. They use approximate an approximate Bayesian computation in a way to give equal representation to the under-represented groups (non-European). They show using a comprehensive evaluation that their method can generate sequences which are close to the real world data, and can be used for downstream supervised tasks like genetic risk scoring.

Pros:
- A simple stochastic model
- Exhaustive evaluation of the generated data across range of metrics for fidelity and diversity
- Model performs well on diversity metrics vs baselines

Cons:
- Model performs lower on the fidelity metrics vs baselines.
- No comparison with baselines on the validity metrics in Section 4.

Question: Does the model do well on diversity metrics because it is trained to represent the under-represented groups in the real data versus the baselines? A discussion on that would be helpful for the readers.

---

### Meta-Review · Program_Chairs · 2022-10-20

**Recommendation:** Accept